# A Greater Increase in Complement C5a Receptor 1 Level at Onset and a Smaller Decrease in Immunoglobulin G Level after Recovery in Severer Coronavirus Disease 2019 Patients: A New Analysis of Existing Data with a New Two-Tailed *t*-Test

**DOI:** 10.3390/biology12091176

**Published:** 2023-08-28

**Authors:** Torao Ishida, Ken Takagi, Guifeng Wang, Nobuyuki Tanahashi, Jun Kawanokuchi, Hisayo Takagi, Yi Guo, Ning Ma

**Affiliations:** 1Project for Advanced Science, Suzuka University of Medical Science, Suzuka 510-0293, Japan; 2College of Traditional Chinese Medicine, Tianjin University of Traditional Chinese Medicine, Tianjin 300193, China

**Keywords:** a greater increase in C5aR1 level at onset, a smaller decrease in IgG level after recovery, severer COVID-19 patients, a new analysis with Ishida’s *t*-test1 and *t*-test2

## Abstract

**Simple Summary:**

It is important to know exactly the difference in changes in Complement C5a Receptor 1 (C5aR1) levels at onset and in Immunoglobulin G (IgG) levels after recovery between severe and non-severe coronavirus disease 2019 (COVID-19) patients to reduce the severity of the disease and prevent reinfection with severe acute respiratory syndrome coronavirus 2. We found that some of these changes in C5aR1 and IgG levels over time were dependent on their initial levels and not suitable for analysis by conventional statistical tests. We developed new t-tests that correctly examine the above changes. Our new t-test suggested a greater increase in C5aR1-levels at onset and a smaller decrease in IgG-levels after recovery in COVID-19 patients than non-COVID-19 patients, which were not detected by conventional statistical tests. Thus, the clinical trials should be analyzed with not only conventional statistical tests but also our new t-test.

**Abstract:**

(1) Background: It is our purpose to identify the differences in the changes in Complement C5a receptor 1 (C5aR1) levels showing the degree of inflammation at onset and Immunoglobulin G (IgG) levels showing the extent of survival of the virus fragments after recovery between coronavirus disease 2019 (COVID-19) and pneumonia coronavirus disease (non-COVID-19) for saving patients’ lives. (2) Methods: First, the studies showing these markers’ levels in individual patients before and after the passage of time were selected from the PubMed Central^®^ databases with the keywords (((COVID-19) AND individual) NOT review) AND C5a/IgG. Then, no changes in these markers’ levels with conventional analyses were selected from the studies. Finally, the no changes were reexamined with our new two-tailed *t*-test using the values on the regression line between initial levels and changed levels instead of the mean or median of changed levels as the expected values of changed levels. (3) Results: Not conventional analyses but our new *t*-test suggested a greater increase in C5aR1-levels at onset and a smaller decrease in IgG-levels after recovery in COVID-19 patients than non-COVID-19 patients. (4) Conclusion: Our new *t*-test also should be used in clinics for COVID-19 patients.

## 1. Introduction

COVID-19 (coronavirus disease 2019) is a new pandemic disease caused by infection with SARS-CoV-2 (severe acute respiratory syndrome coronavirus 2) [1,2,3,4,5,6]. C5a anaphylatoxin and its receptor C5aR1 play a key role in the initiation and maintenance of several inflammatory responses by recruiting and activating neutrophils and monocytes in the lungs [7]. It is important to know exactly the difference in changes in C5aR1 levels at onset and in antibody, especially Immunoglobulin G (IgG), levels after recovery between severe and non-severe COVID-19 patients to reduce the severity of the disease and prevent reinfection with SARS-CoV-2 because C5aR1 levels show the degree of inflammation at onset and IgG levels show the extent of survival of the virus fragments after recovery [8,9,10,11,12,13,14,15,16,17,18,19,20,21,22]. Carvelli et al. [8] reported that an increase in soluble C5a levels proportional to COVID-19 severity and high levels of C5aR1 expression in blood and pulmonary myeloid cells and anti-C5aR1 therapeutic monoclonal antibodies prevented C5a-mediated human myeloid cell recruitment and activation and inhibited acute lung injury in human C5aR1 knockin mice, but there was no statistically significant change in the concentration of C5a desArg (complement C5a removal of the C-terminal arginine) in plasma or % C5aR1 (receptor for C5a)-expressing neutrophils and monocytes between <72 h and days 5–10 after the beginning of hospital care from pneumonia (with two-tailed Wilcoxon signed-rank tests [23]) and ARDS (acute respiratory distress syndrome) patients with the Kenward–Roger method [24,25]. Chen et al. [26] reported that there was a statistically significant change in anti-RBD (receptor binding domain) IgA and neutralization activities, but there was no statistically significant change in the IgM, IgG, and IgA titers specific to the spike protein between hospital discharge and follow-up visit between 21 days and 28 days after discharge among non-severe and severe patients with paired *t*-tests [27] for normal distributions and Mann–Whitney U tests [28] for non-normal distributions. Yang et al. [29] reported that there were no significant changes in the levels of antibody against SARS-CoV-2 surface spike protein RBD in either recurrent-positive or non-recurrent-positive patients between one and two weeks post-discharge. Additionally, there were no significant changes in the levels in recurrent-positive patients between one week before recurrent-positive detection and the time of the detection or between detection time and one week after detection with Mann–Whitney U tests [28].

However, M. Paris et al. [30] reported that combination antiretroviral therapy (cART) did not change the percent of PD-1^high^CTLA-4^low^CD127^high^ early/intermediated CD4^+^ T cells of human immunodeficiency virus (HIV)-infected patients (*n* = 14, *p* = 0.194 with the Wilcoxon signed-rank test [23]) but increased the percent of the marker limited to initial CD4 counts less than 200 (*n* = 9, *p* = 0.0273 with the Wilcoxon signed-rank test [23]). From their report, we hypothesized as follows. (I) There were two kinds of patients. One was the patient whose initial percent of the marker was less than a particular value, and the other was the patient whose initial percent of the marker was greater than the particular value. (II) The treatment increased the percent of the marker of the former and decreased the percent of the marker of the latter. (III) By treating the increase and the decrease in the same way as variation, the Wilcoxon signed-rank test [23] misled R. M. Paris et al., indicating that cART did not increase the percent of the marker with the signed-rank test without separating its increase reaction and decrease reaction. Such a reaction is not detected with the regular regression line between the initial marker values and the marker values after a treatment [31]. 

We thought a similar phenomenon occurred in COVID-19 patients, and then we found that some of these changes in C5aR1 and IgG levels over time (*C*) were dependent on their initial levels (*X*) and not suitable for analysis by conventional statistical tests, such as analysis of covariance [32], analysis of variance [33], Bonferroni’s multiple comparison test [34], Box’s M test [35], the Breusch–Pagan test [36], the Brunner–Munzel test [37], the Cochran–Armitage test [38,39], Cochran’s Q test [40], the Cochran–Mantel–Haenszel test [41,42], the cumulative chi-squared test [43], Dixon’s outlier test [44], Dunnett’s multiple comparison test [45], Friedman’s test [46,47], the F test [48], the G test [49], the generalized Wilcoxon test [50], the Jarque–Bera test [51], the Jonckheere–Terpstra test [52], the Kenward–Roger method [24,25], the Kruskal–Wallis test [53], the Levene test [54], the Lilliefors test [55,56], the linear-by-linear association test [57], the log-rank test [58], the Mann–Whitney U test or Wilcoxon rank sum test [28], the Mantel extension test [59], the McNemar test [60], the paired *t*-test [27], Pearson’s chi-square test [61], Scheffe’s multiple comparison test [62], Shirley–Williams’ multiple comparison test [63,64], Smirnov–Grubbs’ test [65], Steel’s multiple comparison test [66], the Student’s *t*-test [27], Thompson’s rejection test [67], Tukey’s multiple comparison test [68], Welch’s *t*-test [69], the Wilcoxon signed-rank test [23], Williams’ multiple comparison test [70,71], Yates’s correction for continuity [72], and the Z test [73].

We developed new statistical tests (Ishida’s *t*-test1 and *t*-test2) that correctly examine the above changes and then we analyzed the above changes with Ishida’s *t*-test1 and *t*-test2 to obtain different results from their papers as follows.

## 2. Methods

### 2.1. Data Sources

The data source of this meta-analysis was all fields of the PMC databases (up to 1 June 2021).

### 2.2. Selection of the Studies

We searched the literature with the keywords “(((COVID-19) AND individual) NOT review) AND C5a” and “(((COVID-19) AND individual) NOT review) AND IgG”. Then, we selected studies obtained by the keyword search by asking whether they showed individual initial expression levels and the levels after the passage of time of C5a/C5aR1 during onset and antibodies after recovery in patients without significance. A new regression line was used to detect the dynamics where a treatment including time elapsed increases the marker values of particular subjects but decreases those of the others.

(1) A marker value of an individual subject before and after a treatment shall be set as *X* and *Y*, respectively. The changed value (*Y* − *X*) of the marker value after the treatment shall be set as *C*. Thus, *C* = *Y* − *X*. *X* and *C* shall be plotted on a graph with the *X*-axis on the horizontal axis and the *C*-axis on the vertical axis. Positive *C* indicates an increase in the marker value with the treatment. Zero for *C* indicates no change in the marker value with the treatment. Negative *C* indicates a decrease in the marker value with the treatment.

(2) The regression line between *X* and *Y* used regularly [31] does not detect such reactions. Thus, in this study, we used the regression line between *X* and *C* instead of *X* and *Y*. The regression line between *X* and *C* by the method of least squares shall be drawn on the above graph. When the slope (*β*) of the regression line is zero, the regression line crosses only the *C*-axis at the *C*-axis intercept (*α*) but not the *X*-axis. At that time, the regression line will be shown as *E* = *α*. This equation indicates that *C* is independent of *X* when *β* is zero. In this condition, the treatment will only increase or only decrease or not change the marker values of all subjects. When *β* is not zero, the regression line crosses not only the *C*-axis at the *C*-axis intercept (*α*) but also the *X*-axis at the *X*-axis intercept (*γ*). In this time, the regression line will be shown as *E* = *β*(*X* − *γ*), where *E* is the expected value of *C*. This equation indicates that *E* is normally (*β* > 0) or inversely (*β* < 0) dependent on *X*, and *E* is zero at *γ*. When *β* is not zero with statistical significance and the value of *γ* exists between the value of minimum *X* and the value of maximum *X*, the treatment will increase the marker values of particular subjects but decrease those of the others, because *E* at *X* less than *γ* is positive and *E* at *X* greater than *γ* is negative when *β* is negative with statistical significance, and *E* at *X* greater than *γ* is positive and *E* at *X* less than *γ* is negative when *β* is positive with statistical significance. Figure 1 shows a schematic model for the regression lines between X and *C* when *β* is negative (*E*_1_ line) and when *β* is positive (*E*_2_ line).

### 2.3. Data Extraction and Synthesis

*α*, *β*, its *SE*, and its *p*-value of dynamics described in selected studies were obtained with SAS JMP 10 (Corporate Headquarters 100 SAS Campus Drive, Cary, NC, USA). *γ*, *SD*, and *d* of the dynamics were obtained from *γ* = –*α*/*β*, *SD* = *SE*$\sqrt{N}$, and *d* =│*Mc*│/*SD*, respectively, where *N* indicates the number of subjects. The total number of subjects used in this study was 124. The dynamics where *β* was not zero with significance (*p* < 0.05) were extracted from the abovementioned selected studies.

### 2.4. Ishida’s t-Test1

Ishida’s *t*-test1 is a new two-tailed *t*-test fit for paired samples where *β* ≠ 0 with *p* < 0.05*. Mp*; *Mn* and *Me*; and *SD*_1_, *SE*_1_, *t*_1_, and *d*_1_ were calculated using Excel, where the following formulae were incorporated.

*Mp* = $\frac{1}{N}\sum_{i=1}^{Np}\left(Epi\right)$, where *Np* = the number of subjects having *Ep*, *Epi* = *Ep* of particular subject *i*, and *Ep* = positive expected value of *C*.

*Mn* = $\frac{1}{N}\sum_{i=1}^{Nn}\left(Eni\right)$, where *Nn* = the number of subjects having *En*, *Eni* = *En* of particular subject *i*, and *En* = negative expected value of *C*.
$$\mathrm{Me}=\frac{1}{N}\sum_{i=1}^{N}\left(Ei\right)=\frac{1}{N}\sum_{i=1}^{Np}\left(Epi\right)+\frac{1}{N}\sum_{i=1}^{Nn}\left(Eni\right)=\mathrm{Mp}+\mathrm{Mn}.\;{\mathrm{SD}}_{1}={\left[\frac{1}{N-1}\sum_{i=1}^{N}{\left(Ci-Ei\right)}^{2}\right]}^{\frac{1}{2}},\;{\mathrm{SE}}_{1}={\left[\frac{1}{N-1}\sum_{i=1}^{N}{\left(Ci-Ei\right)}^{2}\right]}^{\frac{1}{2}}/\sqrt{N},\;t_{1}\text{-}\mathrm{value}=\left|Me\right|/{\mathrm{SE}}_{1}\;\mathrm{and}\;d_{1}=\left|Me\right|/{\mathrm{SD}}_{1}$$

The *p*_1_-values were determined by inputting the number of degrees of freedom and *t*_1_ into the Excel 2019 T.DIST.2T function. A *p*_1_ of <0.05 was considered statistically significant.

In general, *C* includes the measurement errors due to double measurement called RTM by Galton [74]. Thus, the marker values without the passage of time should be observed as placebo controls for *C*. *C* should be cut off by the values of the placebo controls. When there was no placebo for the changes with the passage of time, we examined the significance of the difference between *C* of the dynamics in patients in any two related groups with Ishida’s *t*-test2, described later, to estimate the significance of *C* under the assumption that at least one group was significantly different from RTM [74] if there was a significant difference between two groups.

### 2.5. Ishida’s t-Test2

Ishida’s *t*-test2 is a new two-tailed *t*-test fit for unpaired samples, where *β* ≠ 0 with *p* < 0.05. *dMe*, *SD*_2_, *SE*_2_, *t*_2_, and *d*_2_ are calculated using Excel, where the following formulae are incorporated.

*dMe* = *Me* of particular group *k* − *Me* of particular group *l.*

*SD*_2_ = ${\left\{\sum_{j=1}^{m}{\left[\sum_{i=1}^{Nj}{\left(Ci-Ei\right)}^{2}\right]}_{j}/\sum_{j=1}^{m}\left(Nj-1\right)\right\}}^{\frac{1}{2}}$, where *m* and *Nj* indicate the number of groups and the number of subjects of a particular group *j*, respectively.
$${\mathrm{SE}}_{2}={\left\{\sum_{j=1}^{m}{\left[\sum_{i=1}^{Nj}{\left(Ci-Ei\right)}^{2}\right]}_{j}/\sum_{j=1}^{m}\left(Nj-1\right)\right\}}^{\frac{1}{2}}*{\left[\frac{1}{Nk}+\frac{1}{Nl}\right]}^{\frac{1}{2}}.\;t_{2}=\left|dMe\right|/{\mathrm{SE}}_{2}\;\mathrm{and}\;d_{2}=\left|dMe\right|/\mathrm{SD}2.$$

When the number of groups was two, the *p*_2_-values were determined by inputting the number of degrees of freedom and *t*_2_ into the Excel 2019 T.DIST.2T function. When the number of groups was three or more than three, the *p*_2_-values were determined by inputting the number of degrees of freedom, the number of groups, and *t*_2_ into the function “>ptukey (*t*-value*sqrt(2), the number of groups, the number of degrees of freedom, lower.tail = FALSE))” of the open software R version 3.4.1. A *p*-value of <0.05 was considered statistically significant.

### 2.6. Validation

R. M. Paris et al. reported that combination antiretroviral therapy (cART) did not change the percent of PD-1^high^CTLA-4^low^CD127^high^ early/intermediated CD4^+^ T cells of human immunodeficiency virus (HIV)-infected patients (*n* = 14, *p* = 0.194 with the Wilcoxon signed-rank test [23]) (hereinafter referred to as b1) but increased the percent of the marker limited to initial CD4 counts less than 200 (*n* = 9, *p* = 0.0273 with the Wilcoxon signed-rank test [23]) (hereinafter referred to as b2) [30]. Generally speaking, sample size should be determined according to (R$\frac{\sigma }{AE}$)^2^, where R, *σ*, and *AE* indicate a constant number determined by a confidence coefficient, the standard deviation of the population, and an allowable error, respectively. And when confidence coefficient is 95%, R is 1.95. However, b2 was a part of b1 and cART increased the percent of the marker (*p* = 0.0273 with the Wilcoxon signed-rank test [23]) and b1 (9 subjects of b2 plus 5 subjects) did not change the percent of the marker (*p* = 0.194 with the Wilcoxon signed-rank test [23]). Thus, cART must decrease the percent of the marker of the rest of the 5 subjects with statistical significance. Thus, 14 of the sample of b1 must be enough for the validation of our methods. Thus, we validated our regression line and statistical test with b1. We estimated *X* (the initial percent of the marker) and *Y* (the percent of the marker after cART) from lines drawn in Figure 2 in their report. We calculated *C* (*X* − *Y*)) (the changed value of the percent of the marker after cART). We calculated the regression line between *X* and *C*, *β* and *γ* of the regression line and their *SD* and *p*-value with SAS JMP 10. A *p*-value of less than 0.05 was considered statistically significant. We also calculated *Mp* and *Mn*, their *SE* and *t*-values with the Excel function incorporating our statistical formula. The *p*-value for *Mp* and *Mn* was determined with the Excel 2010 T.DIST.2T function. A *p*-value of less than 0.05 was considered statistically significant. The results are shown in Figure 2.

According to our test, *β* was negative (−1.346 ± 0.355) with statistical significance (*p* = 0.0026) and *γ* was positive (6.429 ± <3.185) with statistical significance (*p* < 0.0036). Both *Mp* (3.382 ± 1.117%) and *Mn* (−0.910 ± 1.117%) were not zero with statistical significance (*p* = 0.0453). Thus, there were two kinds of patients. One was the patient (b1_a_, *n* = 11) whose initial percent of the marker was less than 6.429%, and the other was the patient (b1_b_, *n* = 3) whose initial percent of the marker was greater than 6.429%. The cART increased the percent of the marker of the former by 3.3822% and decreased the percent of the marker of the latter by 0.910% with statistical significance (*p* = 0.0453). This result was consistent with the following results by separate analyses of b1_a_ and b1_b_ with conventional tests (the paired *t*-test [27] and the Wilcoxon signed-rank test [23]). cART increased the percent of the marker of b1_a_ (*n* = 11) by 4.3040 ± 1.7203% with statistical significance (*p* = 0.026 with the Wilcoxon signed-rank test [23]) and decreased the percent of the marker of b1_b_ (*n* = 3) by 4.2448 ± 0.3445% with statistical significance (*p* = 0.007 with the paired *t*-test [27]). Conventional tests misled R. M. Paris et al. into thinking that cART did not change the percent of the marker of b1.

### 2.7. Risk of Bias in the Methods

There was risk of bias due to limitation of the databases, RTM [74] by double measurements, limitation of the number of subjects whose data were able to be estimated from the spots or lines drawn in figures and estimation errors, and limitation of the validity of Ishida’s *t*-test1 and *t*-test2.

## 3. Results

### 3.1. Selection of Studies

According to the procedure for the selection of studies described in the Methods section and in boxes 1–4 in Figure 3, we selected one study on C5a/C5aR1 [8] and two studies on antibodies [26,29] in COVID-19 patients from 2012 studies in the literature selected from the PMC (PubMed Central^®^) databases with the keywords (((COVID-19) AND individual) NOT review) AND C5a/IgG. We analyzed the *p*-values of slope values (*β*) of 65 regression lines between *X* and *C* made by the method of least squares (*E* line) (Figure 4 and Figure 5) of 39 dynamics and 26 sub-dynamics (Table 1, Table 2 and Table 3) described in the three studies [8,26,29].

### 3.2. Extraction of Dynamics Dependent on X

According to the procedure for the extraction of dynamics described in the Methods section and in the latter half of Figure 3, we extracted eighteen dynamics and twelve sub-dynamics dependent on *X* (Table 1). The *C*-axis intercept (α), slope value (*β*), *X*-axis intercept (*γ* (= −*α/β*)) of the *E* line, standard error (*SE*), standard deviation (*SD* (=*SE*$\sqrt{N}$)), mean of *C* (*Mc*), *p*-value (*p*), and effect size (Cohen’s *d* [75] (*d* (=|*Mc*|/*SD*)) for *β* of the extracted dynamics are shown in the upper left of Table 2. All of the *β* values of the *E* line for 18 dynamics and 12 sub-dynamics were significantly (*p* < 0.05) negative. Thus, in these dynamics, *E* was significantly inversely proportional to *X* to the extent represented by the *p*-value for *β* described in the upper left of Table 2.

### 3.3. Analysis of the Extracted Dynamics with Ishida’s t-Test1

We analyzed positive (*Mp*) and negative (*Mn*) components of the mean (*Me*) of the expected value (*E*) of *C*, *Me*, standard deviation (*SD_1_*), *p*-value (*p*_1_), and effect size (*d*_1_) for the *Me* of the extracted dynamics with Ishida’s *t*-test1 (upper right of Table 2). When *p*_1_ < 0.05, *C* was also significantly inversely nearly proportional to *X* to the extent represented by the *p*-value for *Me* described in the upper right of Table 2. In one dynamic (109) and five sub-dynamics (101s, 109n, 109s, 10yn, and 10ws), the *E* of all subjects decreased with the passage of time from hospital discharge to days 21–28 after hospital discharge. However, in 13 dynamics (081, 091, 092, 093, 094, 101, 107, 108, 10y, 10w, 211, 212, and 221) and 5 sub-dynamics (107n, 108n, 10ys, 10wn, and 10ws), the *E* of those subjects whose *X* values were less than *γ* increased and those of other subjects decreased with the passage of time significantly to the extent represented by the *p*-value for *Me* described in the upper right of Table 2.

Generally, the measuring of values before and after treatment included RTM (regression to the mean) [74]. Therefore, the above *Me* should subtract the *Me* of the placebo control for the passage of time. However, no placebo control for the passage of time was described in any of the 3 studies [8,26,29] or the others (33 studies on C5a/C5aR1 and 1976 studies on antibodies) described in Figure 3. Therefore, we could not obtain *Me* free from RTM [74].

### 3.4. Analysis of the Extracted Dynamics with Ishida’s t-Test2

As we could not obtain *Me* free from RTM [74], we analyzed the difference between the *Me* of a particular group *k* and the *Me* of a particular group *l* (*dMe*) under the assumption that at least one group was significantly different from RTM [74] if there was a significant difference between the two groups.

We analyzed the difference between the *dMe*, standard deviation (*SD*_2_), *p*-value (*p*_2_), and effect size (*d*_2_) for *dMe* of the related two dynamics with Ishida’s *t*-test2 (lower corner of Table 2). The d*Me* of pneumonia (non-COVID-19) (081, 091, and 093) and ARDS (COVID-19) (082, 092, and 094) patients and of non-severe (107n–109n and 10yn) and severe (107s–109s and 10ys) patients was significant to the extent represented by the *p*-value for *dMe* described in Table 2.

There was risk of bias due to limitation of databases, RTM [74] by double measurements, limitation of the number of subjects whose data were able to be estimated from the spots or lines drawn in figures and the estimation errors, and limitation of the validity of Ishida’s *t*-test1 and *t*-test2.

## 4. Discussion

The *E* line crossed the *C*-axis at *α*. When *β* ≠ 0 with *p* < 0.05, this line also crossed the *X*-axis at *γ*. Thus, the equation for this line was described as *E* = *β*(*X* − *γ*). A value obtained from this equation in which *Xi* (*X* of particular subject *i*) was put was set as *Ei*. The deviation of *Ci* (*C* of particular subject *i*) from *Mc* is *Ci* − *Mc.* Thus, that of *Ci* from *Ei* was set as *Ci* − *Ei*. The equation showed that *E* was dependent on *X*. When *β* ≠ 0 with *p* < 0.05 and the *p*-value of *Me* was less than 0.05, *Ci* existed on or near *Ei*. Thus, *C* was also significantly nearly dependent on *X.* The *E* line was made by the method of least squares. Thus,$\;\sum_{i=1}^{N}{\left(Ci-Ei\right)}^{2}$ was the minimum and ≤$\sum_{i=1}^{N}{\left(Ci-Mc\right)}^{2}$. Thus, when *C* was nearly significantly dependent on *X,* the expected value of *C* was not *Mc* but *Ei*. For that reason, Ishida’s *t*-test1 was developed by replacing the *Mc* and [*Ci* − *Mc*] of the paired *t*-test [27] with the *Me* and [*Ci* − *Ei*], and Ishida’s *t*-test2 was developed by replacing the *dMc* (*Mc* of particular group *k* − *Mc* of particular group *l*) and *d*[*Ci* − *Mc*] ([*Ci* − *Mc*] of particular group *k* − [*Ci* − *Mc*] of particular group *l*) of unpaired *t*-tests (Student’s *t*-test [27], Welch’s *t*-test [70], and Tukey’s multiple comparison test [68]) with the *dMe* and *d*[*Ci* − *Ei*] ([*Ci* − *Ei*] of particular group *k* − [*Ci* − *Ei*] of particular group *l*) under the condition that *C* was significantly nearly dependent on *X*. The following was demonstrated using 092 dynamics as an example.

Figure 4 shows the *E* line for 092 (dynamics of % C5aR1-expressing neutrophils in ARDS (COVID-19) patients between <72 h and days 5–10 after the beginning of hospital care shown in extended Figure 2 by Carvelli et al. [8]).

The *β* for 092 was significantly negative (*p* = 0.0002), and *p_1_* for *Me* was 0.00004. Figure 4 shows that *C* existed on or near by the *E* line and far from the *Mc* (compare the difference between “the red closed circle sign” and “+ sign” and the difference between “the red closed circle sign” and “x sign” in Figure 4). Thus,$\;\sum_{i=1}^{N}{\left(Ci-Ei\right)}^{2}$ < $\sum_{i=1}^{N}{\left(Ci-Mc\right)}^{2}$ in 092. Thus, the expected value of *Ci* was *Ei* but not *Mc*, and the expected value of *Mc* was *Me* because the *C* of 092 was significantly nearly dependent on *X*.

As previously mentioned, measuring values before and after the passage of time included RTM [74], but the placebo control for the passage of time was not described in the 2012 studies on C5a/C5aR1 and antibodies selected from the PMC databases. We should plan clinical trials including placebo controls for the passage of time, which could be easily obtained by measuring marker values again immediately after measuring their initial values.

Unlike the conclusions of Carvelli et al. [8], the analysis of their paper with Ishida’s tests suggested the following under the assumption that at least one group was significantly different from RTM [74], as there was a significant difference between the two groups. (1) C5a levels of pneumonia (non-COVID-19) patients decreased significantly (*p* = 0.004), with a large effect size (*d* = 0.77), but C5a levels of ARDS (COVID-19) patients increased without significance (*p* = 0.068) and with a moderate effect size (*d* = 0.38) for at least 10 days after the beginning of hospital care. (2) There was a significant (*p* = 0.002) difference in changes in C5a levels between pneumonia (non-COVID-19) patients and ARDS (COVID-19) patients for the passage of time mentioned above, with a large effect size (*d* = 1.01). (3) The % C5aR1-expressing neutrophils of pneumonia (non-COVID-19) patients increased slightly but significantly (*p* < 0.0001), with a large effect size (*d* = 9.78), but those of ARDS (COVID-19) patients increased ten times more than those of pneumonia (non-COVID-19) patients significantly (*p* < 0.0001), with a large effect size (*d* = 8.90), for the passage of time mentioned above. (4) There was a significant (*p* < 0.0001) difference in the % C5aR1-expressing neutrophils between pneumonia (non-COVID-19) patients and ARDS (COVID-19) patients at the passage of time mentioned above, with a large effect size (*d* = 11.4). (5) The % C5aR1-expressing monocytes of pneumonia (non-COVID-19) patients increased significantly (*p* = 0.0002), with a large effect size (*d* = 1.29), but those of ARDS (COVID-19) patients increased two-and-a-half times more than those of pneumonia (non-COVID-19) patients significantly (*p* < 0.0001), with a large effect size (*d* = 3.13), at the passage of time mentioned above. (6) There was a significant (*p* < 0.0001) difference in the % C5aR1-expressing monocytes between pneumonia (non-COVID-19) patients and ARDS (COVID-19) patients at the passage of time mentioned above, with a large effect size (*d* = 1.87).

Unlike the conclusions of Chen et al. [26], the analysis of their paper with Ishida’s *t*-test1 suggested that not only anti-RBD IgA (109) but also other antibodies (anti-RBD IgM (101), anti-S1 IgG (107), anti-NP (nucleoprotein) IgG (108), and anti-S1 IgA (10y)) reduced with the passage of time by each *Me* with each *p*-value described in the upper right of Table 2, and these reduced antibodies might have been responsible for the declining trend of neutralizing activities (10w) by 1973 titers (*p* < 0.0001 with *d* = 3.10). Analysis with Ishida’s *t*-test2 suggested that the anti-S1 IgG (107) and anti-NP IgG (108) of non-severe patients decreased more than those of severe patients (*p* < 0.0001 with *d* = 4.65 and *p* < 0.0001 with *d* = 8.23), respectively, under the same assumption described above.

There were no significant differences in antibody levels between 211 and 212, between 221 and 222, and between 229 and 22x with Ishida’s *t*-test2. Our results could not be differentiated from the conclusion of Yang et al. [29] because we could not assume that at least one of the two groups compared was significantly different from RTM [74].

## 5. Conclusions

When clinical trials observing the effects of the passage of time are planned, placebo controls for the passage of time and analysis of *β* of the regression line between *X* and *C* should be included. When *β* ≠ 0 with *p* < 0.05, the clinical trials should be analyzed with Ishida’s *t*-test1 and *t*-test2.

## Figures and Tables

**Figure 1 biology-12-01176-f001:**
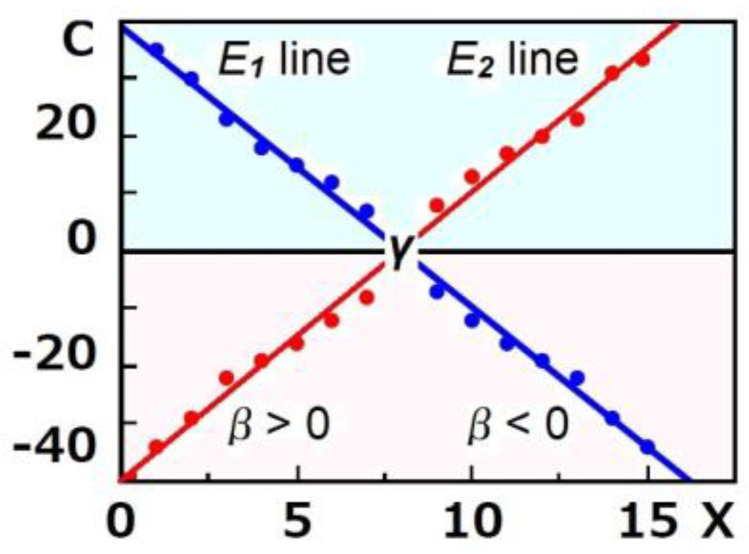
A schematic model for the regression lines between *X* and *C* when *β* is negative (*E*_1_ line, blue) and when *β* is positive (*E*_2_ line, red). Light−sky−blue zone indicates expected increase area. Light−pink zone indicates expected decrease area.

**Figure 2 biology-12-01176-f002:**
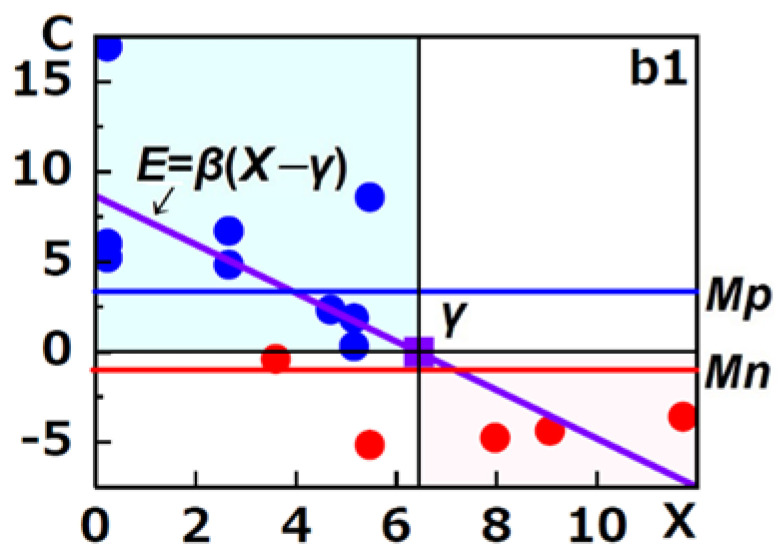
*E* line for b1. The purple diagonal line, the blue line, and the red line indicate the regression line *E* = *β*(*X* − *γ*) between *X* and *C* of b1, the Mp line, and the Mn line, respectively. The purple square sign, the blue closed circle sign, and the red closed circle sign indicate *X*-axis intercept value (*γ*), positive *C*, and negative *C*, respectively. The light-sky−blue zone and the light−pink zone indicated the expected increase area and the expected decrease area, respectively. The *X*, *C*, *E*, *β*, *γ*, *Mp*, and *Mn* are described in the text.

**Figure 3 biology-12-01176-f003:**
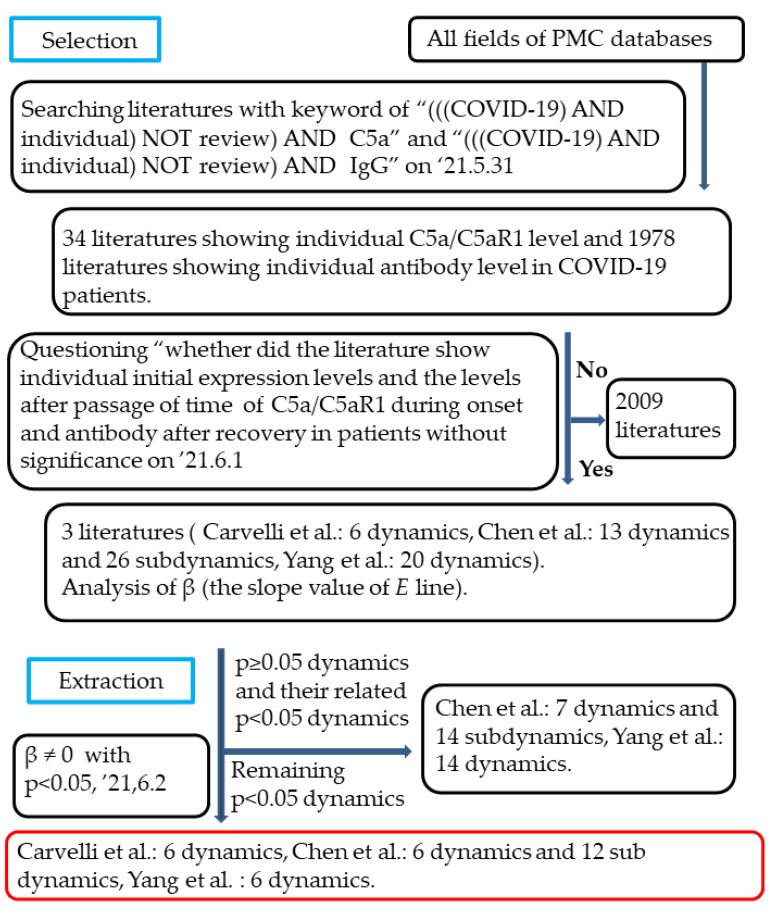
Procedures for study selection and extraction of C5a/C5aR1 or antibody dynamics in COVID-19 patients [8,26,29].

**Figure 4 biology-12-01176-f004:**
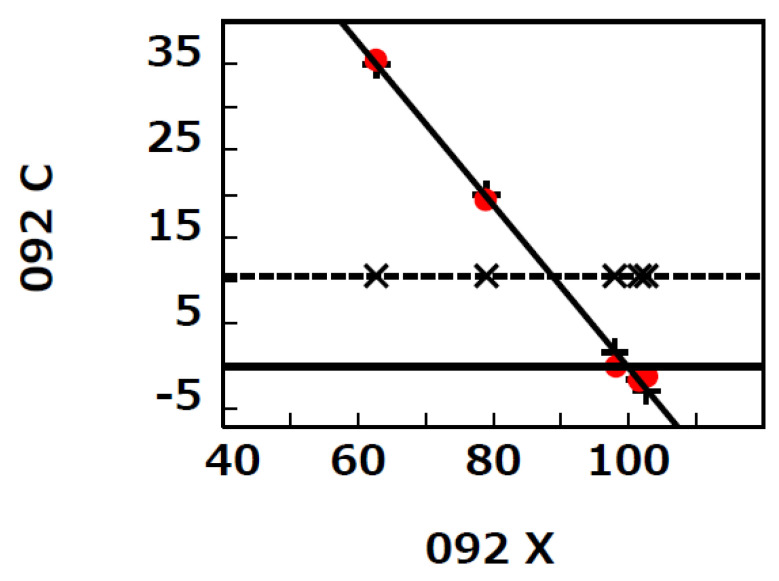
*E* line for 092. The solid diagonal line indicates the *E* line in which the *p*-value of *β* was <0.05. The dotted horizontal line indicates the *Mc* line in which the *p*-value of *Mc* was ≥0.05. The red closed circle sign, + sign, and x sign indicate the points at (*Xi*, *Ci*), (*Xi*, *Ei*), and (*Xi*, *Mc*), respectively. *E* line, 092, *β*, *Xi*, *Ci*, *Ei*, and *Mc* are described in the text.

**Figure 5 biology-12-01176-f005:**
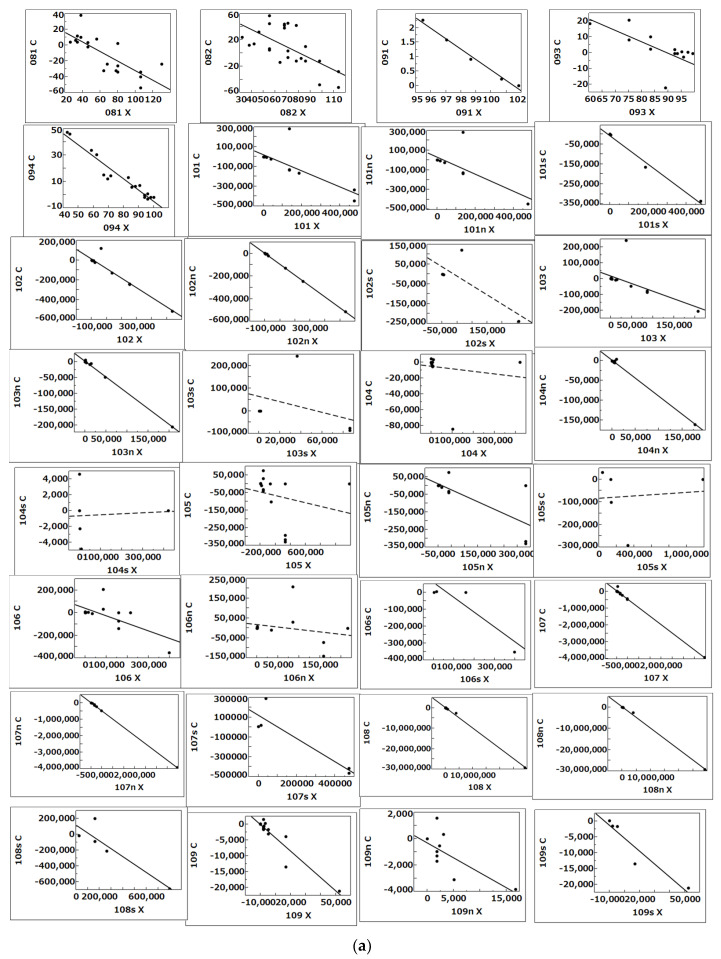
(**a**) *E* lines for 081–109s. Contents of the dynamics with their sources are described in Table 1 or Table 3. Solid and dotted lines indicate that the *p*-value of the slope of the *E* line was <0.05 and ≥0.05, respectively. (**b**) *E* lines for 10x–22x. Contents of the dynamics with their sources are described in Table 1 or Table 3. Solid and dotted lines indicated that the *p*-value of the slope of the *E* line was <0.05 and ≥0.05, respectively.

**Table 1 biology-12-01176-t001:** Contents of the included dynamics with their sources.

DYN Name	Contents and Sources of Dynamics (DYN)
081, 082	Dynamics of concentration (ng/mL) of C5a desArg in plasma from pneumonia (non-COVID-19) (green: 081, *n* = 19) and ARDS (COVID-19) (red: 082, *n* = 25) patients between T0 and T1in extended Figure 1 by Carvelli et al. [8]
091 092	Dynamics of % C5aR1-expressing neutrophils in pneumonia (non-COVID-19) (green: 091, *n* = 5) and ARDS (COVID-19) (red: 092, *n* = 5) patients between T0 and T1 in extended Figure 2 by Carvelli et al. [8]
093 094	Dynamics of % C5aR1-expressing monocytes in pneumonia (non-COVID-19) (green: 093, *n* = 15) and ARDS (COVID-19) (red: 094, *n* = 17) patients between T0 and T1 in extended Figure 2 by Carvelli et al. [8]
101	Dynamics of the IgM titer specific to RBD from non-severe (101n, *n* = 15) and severe (101s, *n* = 5) patients between T3 and T4 in Figure 6 (anti-RBD IgM) by Chen et al. [26]
107	Dynamics of the IgG titer specific to S1 from non-severe (107n, *n* = 11) and severe (107s, *n* = 5) patients between T3 and T4 in Figure 6 (anti-S1 IgG) by Chen et al. [26]
108	Dynamics of the IgG titer specific to NP from non-severe (108n, *n* = 11) and severe (108s, *n* = 5) patients between T3 and T4 in Figure 6 (anti-NP IgG) by Chen et al. [26]
109	Dynamics of the IgA titer specific to RBD from non-severe (109n, *n* = 10) and severe (109s, *n* = 5) patients between T3 and T4 in Figure 6 (anti-RBD IgA) by Chen et al. [26]
10y	Dynamics of the IgA titer specific to S1 from non-severe (10yn, *n* = 11) and severe (10ys, *n* = 5) patients between T3 and T4 in Figure 6 (anti-S1 IgA) by Chen et al. [26]
10w	Dynamics of the NAb titer (IC50) from non-severe (10wn, *n* = 11) and severe (10ws, *n* = 5) patients between T3 and T4 described in Figure 6 (Nab) by Chen et al. [26]
211 212	Dynamics of IgM levels specific to RBD (titers) from non-recurrent-positive (211, *n* = 35) and recurrent-positive (212, *n* = 6) patients between T5 and T6 described in Figure 4 (IgM levels) by Yang et al. [29]
221 222	Dynamics of IgM levels specific to RBD (titers) from recurrent-positive patients between T7 and T8 (221, *n* = 27) and between T8 and T9 (222, *n* = 23) described in Figure 4 (IgM levels) by Yang et al. [29]
229 22x	Dynamics of NAb levels specific to RBD (titers) from recurrent-positive patients between T7 and T8 (229, *n* = 21) and between T8 and T9 (22x, *n* = 18) described in Figure 4 (Nab levels) by Yang et al. [29]

*n*: number of subjects; T0: <72 h after the beginning of hospital care; T1: days 5–10 after the beginning of hospital care; T3: the time point of hospital discharge; T4: days 21–28 after hospital discharge; T5: week 1 hospital discharge; T6: week 2 hospital discharge; T7: one week before recurrent-positive detection; T8: the time point of recurrent-positive detection; T9: one week after recurrent-positive detection; ARDS: acute respiratory distress syndrome; C5a des-Arg: Complement C5a removal of the C-terminal arginine; C5aR1: receptor for the C5a anaphylatoxin; S1, NP, and RBD: S1 protein, nucleoprotein, and receptor binding domain of the spike proteins of SARS-CoV-2; Ig: Immunoglobulin; NAb: neutralizing antibody.

**Table 2 biology-12-01176-t002:** Analysis of the included dynamics in three studies [8,26,29] with regression line and Ishida’s *t*-test1 and *t*-test2.

Grp	*β*	*γ*	*SD*	*p*	*d*	*Mp*	*Mn*	*Me*	*SD* _1_	*p* _1_	*d* _1_
081	−0.61	47.0	0.51	<0.0001	1.19	2.60	−13.91	−11.30	14.77	0.004	0.77
082	−0.87	81.1	1.11	0.0007	0.79	12.64	−3.89	8.75	22.88	0.068	0.38
091	−0.36	101.5	0.05	0.0005	7.12	1.02	−0.02	1	0.10	<0.0001	9.78
092	−0.94	99.9	0.09	0.0002	10.9	11.29	−0.87	10.42	1.17	<0.0001	8.90
093	−0.65	94.6	0.40	<0.0001	1.61	5.73	−0.41	5.32	4.11	0.0002	1.29
094	−0.86	94.1	0.24	<0.0001	3.62	13.73	−0.67	13.06	4.17	<0.0001	3.13
101	−0.81	3 × 10^4^	0.71	0.0007	1.15	8381	−8 × 10^4^	−7 × 10^4^	1 × 10^5^	0.017	0.70
101n	−0.87	4 × 10^4^	0.98	0.0162	0.89	1 × 10^4^	−7 × 10^4^	−5 × 10^4^	1 × 10^5^	0.174	0.44
101s	−0.72	−9583	0.11	0.0053	6.87	0	−1 × 10^5^	−1 × 10^5^	2 × 10^4^	0.0006	7.68
107	−1.01	4 × 10^4^	0.09	<0.0001	11.3	2 × 10^4^	−4 × 10^5^	−3 × 10^5^	8 × 10^4^	<0.0001	4.25
107n	−1.00	1 × 10^4^	0.02	<0.0001	60.9	5310	−5 × 10^5^	−4 × 10^5^	2 × 10^4^	<0.0001	25.77
107s	−1.18	1 × 10^5^	0.68	0.0306	1.73	6 × 10^4^	−2 × 10^5^	−1 × 10^5^	1 × 10^5^	0.128	0.86
108	−1.00	2 × 10^5^	0.05	<0.0001	19.5	7 × 10^4^	−2 × 10^6^	−2 × 10^6^	4 × 10^5^	<0.0001	6.13
108n	−1.00	2 × 10^5^	0.05	<0.0001	18.9	1 × 10^5^	−3 × 10^6^	−3 × 10^6^	4 × 10^5^	<0.0001	7.38
108s	−1.00	1 × 10^5^	0.60	0.0333	1.67	2 × 10^4^	−2 × 10^5^	−2 × 10^5^	1 × 10^5^	0.059	1.17
109	−0.42	−306	0.17	<0.0001	2.47	0	−3260	−3260	2125	<0.0001	1.53
109n	−0.23	−1346	0.27	0.0303	0.83	0	−1127	−1127	1167	0.014	0.97
109s	−0.41	−3275	0.18	0.0139	2.32	0	−7526	−7526	2960	0.005	2.54
10y	−0.82	90.2	0.28	<0.0001	2.96	8.24	−2 × 10^4^	−2 × 10^4^	9263	<0.0001	1.90
10yn	−0.70	−52.0	0.28	<0.0001	2.55	0	−1 × 10^4^	−1 × 10^4^	8674	0.0008	1.43
10ys	−1.01	1303	0.03	<0.0001	39.7	228	−3 × 10^4^	−3 × 10^4^	781	<0.0001	37.35
10w	−0.94	546	0.13	<0.0001	7.28	120	−2093	−1973	637	<0.0001	3.10
10wn	−0.94	679	0.13	<0.0001	7.27	254	−2380	−2126	748	<0.0001	2.84
10ws	−0.93	255	0.04	<0.0001	25.2	0	−1636	−1636	42	<0.0001	38.92
211	−0.26	0.87	0.16	<0.0001	1.66	0.04	−0.56	−0.53	0.57	<0.0001	0.92
212	−0.34	0.72	0.17	0.0073	2.06	0.06	−0.72	−0.67	0.36	0.006	1.85
221	−0.23	0.33	0.11	<0.0001	2.14	0.01	−0.12	−0.11	0.11	<0.0001	1.04
222	−0.45	0.94	0.77	0.011	0.58	0.22	−0.13	0.10	0.82	0.579	0.12
229	−0.27	46.1	0.30	0.0047	0.70	5.33	−18.13	−12.80	57.92	0.323	0.22
22x	−0.80	69.57	0.29	<0.0001	2.78	38.65	−53.97	−15.32	70.12	0.367	0.22
Gr k	Gr l	*dMe*	*SD_2_*	*p* _2_	*d* _2_	Gr k	Gr l	*dMe*	*SD_2_*	*p* _2_	*d* _2_
081	082	−20.1	19.8	0.002	1.01	109n	109s	6399	1908	<0.0001	3.35
091	092	−9.42	0.83	<0.0001	11.4	10yn	10ys	2 × 10^4^	7343	0.0008	2.72
093	094	−7.74	4.14	<0.0001	1.87	10wn	10ws	−489	633	0.173	0.77
101n	101s	7 × 10^4^	1 × 10^5^	0.276	0.66	211	212	0.14	0.55	0.559	0.26
107n	107s	−3 × 10^5^	7 × 10^4^	<0.0001	4.65	221	222	−0.21	0.56	0.198	0.37
108n	108s	−3 × 10^6^	4 × 10^5^	<0.0001	8.23	229	22x	2.52	63.8	0.903	0.04

Abbreviations: *C*-axis intercept (*α*), slope value (*β*), *X*-axis intercept (*γ* (= −*α/β*)) of the regression line between the initial (*X*) and changed (*C*) value with treatment of a marker made by the method of least squares (*E* line); positive (*Mp*) and negative (*Mn*) comportment of mean of expected value of *C* (*Me*); difference between *Me* of group (Grp) k and group (Grp) l (*dMe*); standard deviation and *p*-value of conventional (*SD*, *p*), Ishida’s *t*-test1 (*SD*_1_, *p*_1_), *t*-test2 (*SD*_2_, *p*_2_); |*Mc* (mean of *C*)|/*SD* (*d*), |*Me*|/*SD*_1_ (*d*_1_), |*dMe*|/*SD*_2_ (*d*_2_); contents and sources of 081 − 22x are described in Table 1. Units of *Mp*, *Mn*, *Me*, and *dMe*: ng/mL (081, 082), % (091, 092, 093, 094), titers for IC50 (10w, 10wn, 10ws, 229, 22x), titers (others).

**Table 3 biology-12-01176-t003:** Contents of the excluded dynamics with their sources.

DYN Name	Contents and Sources of Dynamics (DYN)
102	Dynamics of the IgM titer specific to ECD from non-severe (102n, *n* = 11) and severe (102s, *n* = 5) patients between T3 and T4 in Figure 6 (anti-ECD IgM) by Chen et al. [26]
103	Dynamics of the IgM titer specific to S1 from non-severe (103n, *n* = 11) and severe (103s, *n* = 5) patients between T3 and T4 in Figure 6 (anti-S1 IgM) by Chen et al. [26]
104	Dynamics of the IgM titer specific to NP from non-severe (104n, *n* = 11) and severe (104s, *n* = 5) patients between T3 and T4 in Figure 6 (anti-NP IgM) by Chen et al. [26]
105	Dynamics of the IgG titer specific to RBD from non-severe (105n, *n* = 11) and severe (105s, *n* = 5) patients between T3 and T4 in Figure 6 (anti-RBD IgG) by Chen et al. [26]
106	Dynamics of the IgG titer specific to ECD from non-severe (106n, *n* = 11) and severe (106s, *n* = 5) patients between T3 and T4 in Figure 6 (anti-ECD IgG) by Chen et al. [26]
10x	Dynamics of the IgA titer specific to ECD from non-severe (10xn, *n* = 9) and severe (10xs, *n* = 5) patients between T3 and T4 in Figure 6 (anti-ECD IgA) by Chen et al. [26]
10z	Dynamics of the IgA titer specific to NP from non-severe (10zn, *n* = 11) and severe (10zs, *n* = 5) patients between T3 and T4 in Figure 6 (anti-NP IgA) by Chen et al. [26]
213 214	Dynamics of IgG levels specific to RBD (titers) from non-recurrent-positive (213, *n* = 32) and recurrent-positive (214, *n* = 6) patients between T5 and T6 described in Figure 4 (IgG levels) by Yang et al. [29]
215 216	Dynamics of IgA levels specific to RBD (titers) from non-recurrent-positive (215, *n* = 32) and recurrent-positive (216, *n* = 5) patients between T5 and T6 described in Figure 4 (IgA levels) by Yang et al. [29]
217 218	Dynamics of all Ig levels specific to RBD (titers) from non-recurrent-positive (217, *n* = 33) and recurrent-positive (218, *n* = 6) patients between T5 and T6 described in Figure 4 (all Ig levels) by Yang et al. [29]
219 21x	Dynamics of Nab levels specific to RBD (titers) from non-recurrent-positive (217, *n* = 33) and recurrent-positive (218, *n* = 6) patients between T5 and T6 described in Figure 4 (Nab levels) by Yang et al. [29]
223 224	Dynamics of IgG levels specific to RBD (titers) from recurrent-positive patients between T7 and T8 (223, *n* = 27) and between T8 and T9 (224, *n* = 23) described in Figure 4 (IgG levels) by Yang et al. [29]
225 226	Dynamics of IgA levels specific to RBD (titers) from recurrent-positive patients between T7 and T8 (225, *n* = 21) and between T8 and T9 (226, *n* = 20) described in Figure 4 (IgA levels) by Yang et al. [29]
227 228	Dynamics of all Ig levels specific to RBD (titers) from recurrent-positive patients between T7 and T8 (227, *n* = 27) and between T8 and T9 (228, *n* = 25) described in Figure 4 (all Ig levels) by Yang et al. [29]

Each abbreviation is described in Table 1.

## Data Availability

The authors declare that all data supporting the findings of this study are available within the article and in public repositories (doi:10.1038/s41586-020-2600-6 (accessed on 1st June 2021), https://doi.org/10.1371/journal.ppat.1008796 (accessed on 1 June 2021), https://doi.org/10.1080/22221751.2020.1837018 (accessed on 1 June 2021)) or from the corresponding author upon reasonable request.

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
