# Peer review of "A Greater Increase in Complement C5a Receptor 1 Level at Onset and a Smaller Decrease in Immunoglobulin G Level after Recovery in Severer Coronavirus Disease 2019 Patients: A New Analysis of Existing Data with a New Two-Tailed t-Test"

_biology, 2023, doi:10.3390/biology12091176_

Round 1

Reviewer 1 Report

The authors presented here a new analysis of existing data with Ishida’s t-test (a new two-tailed t-test).

The paper look good and the readability of the manuscript is really good and easy to follow. However, the abstract section is quite confusing, seems that words got mixed up. Strongly advise the authors to revise the abstract section.

Moreover, could the authors add a section that show when and how this new t-test (Ishida’s t-test) has been validated

The various sections of the paper are well written, ideas well organized and easy to follow the objective of the paper

The manuscript is well written, and the English level is good enough to understand the content. However, as i stated in the previous section, i think the content of the abstract has been mixed up to the point that in some sentences, the punctuation do not make sens.

Author Response

To Prof. Dr. Reviewer 1

we send our revised revised manuscript in which the revised parts according to your comments are marked with a yellow marker and the revised parts according to other reviewer' comments are marked with sky blue marker for you to understand our intent to revise.

Please see the attachment 2.

Very truly yours. 

Torao Ishida 

Reviewer 2 Report

1-The title should changed and not contain abbreviations and i suggest to remove with Ishida’s t-test1 and t-test2 from it.

2-Funding statement should move to the end of the paper.

3-the literature should be increased.

4- the motivation should be improved.

5-The size of the font is inconsistent.

6- text in Figure 1 should be written by the same font.

7- all equations should be ended by . or ,

Really, the paper is written bad and it is not acceptable to publish in this reputable journal.

Not nice.

Author Response

Prof. Dr. Reviewer 2.

We our revised manuscript in which the reviced parts according to your comment are marked with yellow marker and the revised parts according to other reviewer's comments are marked with sky blue marker for you to understand our intent to revise.  Please see the attachment

Very truly yours

Torao Ishida.

Reviewer 3 Report

This is an interesting review by Torao Ishida et.al, using previously reported data with new analysis using Ishida’s test to determine C5aR1 levels and Immunoglobulin (IgG) levels. Authors also proposed using Ishida’s t-test1 and t-test2 during clinical trials to interpret the data. The authors need to improve the writing and explanation of figures with references. Comments must be improved as noted below.

Lane 16-17: Rewrite “2019” patients to reduce the severity of the disease

Lane 34-36: explanation of relation between C5aR and IgG levels” was missing.

Lane 60: Organ.” means organization?

Lane 63: total how many subjects/patients data were used in this study?

Lane 142: box 4: What is meant by “34 and 1978 literatures”?

Lane 142: box 4: should be “level in Covid-19”

Lane 148: rewrite “first half of Figure 1”

Lane 155: Table 1: what are “081, 082” and label the “columns”

Lane 280, 284 and 287: Remove “extended”

It would have been better to know the pattern of C5aR and IgG levels in “non Covid-19” patients and interpret to support current conclusions.

English writing can be improved

Author Response

Prof. Dr. Reviewer 3.

WE will send our revised manuscript in which the revised parts according to your comment are marked with yellowmarker and the revised parts according to other reviewer's comments are marked with sky blue marker for you to understand our intent to revise. now we attach our revised manuscript.

Very truly yours, 

Torao Ishida
